# LS-APC v1.0: A tuning-free method for the linear inverse problem and its application to source term determination

Ondřej Tichý[1], Václav Šmídl[1], Radek Hofman[1], and Andreas Stohl[2]

[1]Institute of Information Theory and Automation, Czech Academy of Sciences, Prague, Czech Republic
[2]NILU: Norwegian Institute for Air Research, Kjeller, Norway

*Correspondence to:* O. Tichý (otichy@utia.cas.cz)

**Abstract.** Estimation of pollutant releases into the atmosphere is an important problem in the environmental sciences. It is typically formalized as an inverse problem using a linear model that can explain observable quantities (e.g. concentrations or deposition values) as a product of the source-receptor sensitivity (SRS) matrix obtained from an atmospheric transport model multiplied by the unknown source term vector. Since this problem is typically ill-posed, current state-of-the-art methods are based on regularization of the problem and solution of a formulated optimization problem. This procedure depends on manual settings of uncertainties that are often very poorly quantified, effectively making them tuning parameters. We formulate a probabilistic model, that has the same maximum likelihood solution as the conventional method using pre-specified uncertainties. Replacement of the maximum likelihood solution by full Bayesian estimation allows to estimate also all tuning parameters from the measurements. The estimation procedure is based on the Variational Bayes approximation which is evaluated by an iterative algorithm. The resulting method is thus very similar to the conventional approach, but with the possibility to estimate also all tuning parameters from the observations. The proposed algorithm is tested and compared with the standard methods on data from the European Tracer Experiment (ETEX) where advantages of the new method are demonstrated. A MATLAB implementation of the proposed algorithm is available for download.

## 1 Introduction

Estimating the emissions of a substance into the atmosphere can be done in two alternative ways: The first method, a bottom-up approach, is based on a (statistical) model describing the emission process. For greenhouse gases or air pollutants, this is typically based on detailed country statistics (e.g., about energy use) and some proxy information (e.g., population distribution) to spatially disaggregate the emissions. The other method, often called top-down approach (Nisbet and Weiss, 2010), makes use of ambient measurements of the released substance (e.g., atmospheric concentrations) and a model for describing the behavior of the substance in the atmosphere. The emissions are then constrained to values that are compatible with the measured data. The two approaches are complementary, where the top-down approach can be used to verify bottom-up estimates, to identify problems in bottom-up emission inventories, or to improve these inventories (e.g., Lunt et al., 2015). For determining the emissions of greenhouse gases into the atmosphere, such an approach has become very common. In particular, total greenhouse gas emissions are the result of both anthropogenic and natural emissions. Bottom-up inventories for anthropogenic emissions

should, at least in principle, be quite accurate but a verification using top-down methods is desirable (Stohl et al., 2009; Bergamaschi et al., 2015). Natural emissions are often poorly constrained with bottom-up methods and thus the role of top-down methods is even more important (Tans et al., 1990; Rayner et al., 1999).

For other emissions into the atmosphere, such as releases by nuclear accidents (Davoine and Bocquet, 2007; Stohl et al., 2012), nuclear explosions (Issartel and Baverel, 2003), or for emissions of volcanic ash during volcanic eruptions (Kristiansen et al., 2010; Stohl et al., 2011), the problem is very different. While the emission location is often known and sometimes the emission period can be estimated, other bottom-up information on the magnitude of the emissions, their temporal variations and, occasionally, the emission altitude, can be very incomplete or, especially in real time, inexistent. In these cases, emission estimates based on the top-down approach are often the only way to constrain the so-called source term, which quantifies the emissions of a point source as a function of time and, sometimes, altitude. Still, the source term is one of the largest source of errors in modeling and predicting the pollutant dispersion in the atmosphere (Stohl et al., 2012). Since it is key information for decision making in emergency response situations, any improvement of the reliability of the source term estimation is important.

The common approach for source term determination is to combine data measured in the environment (e.g., radionuclide concentrations downwind of the release site) with an atmospheric transport model in a top-down approach. Agreement between a model with calculated source term and measurements can be modeled and optimized using various parameter estimating methods including the Bayesian approach (Bocquet, 2008), maximum entropy principle (Bocquet, 2005b), or cost function optimization (Eckhardt et al., 2008). For computational reasons, this problem is typically formulated as a variant of linear regression. The vector of measurements is assumed to be explained using a linear model with a known source-receptor-sensitivity (SRS) matrix determined using an atmospheric dispersion model (Seibert and Frank, 2004) and an unknown source term vector. Simple solution via the ordinary least-squares method typically yields a poor solution because the problem is often only partially determined and ill-constrained by the available measurement data. Many regularization schemes taking into account physically plausible ranges of parameter values such as non-negativity of the emissions, or other a priori information, for instance on the duration of release, have been proposed providing more realistic solutions. However, especially if the a priori information is incomplete, the regularization terms can also contain tuning parameters which are often selected manually and subjectively. The solution is subsequently highly sensitive to their choice. Therefore, many authors proposed inversion schemes to reduce the dependency on these parameters. Davoine and Bocquet (2007) formulated the inversion problem as minimization problem with Tikhonov regularization term. Similar model was used by Winiarek et al. (2012) where covariance matrices of both, observation errors and source term, are assume to be diagonal with identical elements on each diagonal. The positivity of the source term is enforced using truncation of negative estimates. Three estimation methods were studied to infer model parameters: L-curve, Desroziers' scheme (Desroziers et al., 2005), and brute force using maximum likelihood screening. Diagonal matrices with different diagonal entries were considered in work of Michalak et al. (2005) where maximum likelihood method was used to infer the model parameters. The model was extended by Berchet et al. (2013) where full covariance matrices were considered. Desroziers' scheme and maximum likelihood were used; however, heuristics need to be used due to divergence of the algorithm after a few iterations. To cope better with full covariance matrix of measurements, Ganesan

et al. (2014) follow the work of Michalak et al. (2005) and propose a model for non-diagonal entries using exponential decay with common autocorrelation timescale parameter weighted by estimated diagonal entries. Similar model was then used by Henne et al. (2016) for both covariance matrices, measurement and source term; however, with fixed common autocorrelation timescale parameter for non-diagonal entries. In this paper, we propose a probabilistic model that estimates such tuning parameters from the data using a Bayesian approach with hierarchical prior.

Most of the existing regularization techniques are based on restricted structure of the prior covariance matrix. Various covariance structures for linear models have been studied extensively in the statistical literature, see, e.g., reviews (Pourahmadi et al., 2011; Daniels, 2005). For example, a model of only diagonal structure of the covariance matrix has been proposed to favor sparse solutions (Tipping, 2001). It is possible to use more complex models of the covariance structure using Cholesky decomposition (Pourahmadi, 2000), its modifications (Daniels and Pourahmadi, 2002), or more general decompositions (Khare et al., 2011). The inference mechanism is usually a variant of Monte Carlo simulations. In this work, we choose the prior covariance structure to have two main diagonals in modified Cholesky decomposition. The inference of the posterior is achieved using the Variational Bayes method (Šmídl and Quinn, 2006) which is closely related to algorithms used in this application domain.

We will illustrate the proposed approach in comparison with the commonly used method of Seibert (2001); Eckhardt et al. (2008) and with other state-of-the-art algorithms (Martinez-Camara et al., 2014), least absolute shrinkage and selection operator (LASSO) (Tibshirani, 1996), and the conventional Tikhonov regularization (Golub et al., 1999). We will show how the formal Bayesian approach yields an iterative algorithm closely related to that of (Eckhardt et al., 2008). Many heuristic steps in determining regularization parameters will be replaced by statistical estimates. The most significant advantage over the reference approach is estimation of the tuning parameters from the data. In effect, the proposed algorithm works without manual intervention. The only entries of the proposed algorithm are the vector of measurements and the SRS matrix calculated with a dispersion model. The MATLAB implementation of the derived algorithm is freely available for download. The resulting algorithm is tested and compared using real data from the European Tracer Experiment (ETEX).

## 2  Background

In this Section, we review the standard methodology, as described e.g. by Eckhardt et al. (2008), which is commonly used in source term determination and the Variational Bayes method (Šmídl and Quinn, 2006; Miskin, 2000) which is the key methodology of this work.

### 2.1  Standard Methodology

We choose the work of Eckhardt et al. (2008) as a reference. It is only an example from a family of optimization methods based on linear regression such as (Seibert, 2000), (Seibert, 2001), (Bocquet, 2005b), (Bocquet, 2008), or (Tarantola, 2005). The regularization is achieved by formulating a prior knowledge on the solution and using an iterative algorithm for removing physically unrealistic values in the posterior solution. The basic inverse problem is formulated based on the following linear

model,

$$\mathbf{y} = M\mathbf{x}, \tag{1}$$

where $\mathbf{y}$ is the vector of measurements (typically observed concentrations or, sometimes, deposition values), $M$ is a known source-receptor-sensitivity (SRS) matrix (Seibert and Frank, 2004), and $\mathbf{x}$ is the unknown source term vector. Solution of the problem via the ordinary least-squares method is not feasible since matrix $M$ is typically ill-conditioned.

Regularization of the problem proposed in (Eckhardt et al., 2008) is based on minimization of the cost function $J = J_1 + J_2 + J_3$:

$$J_1 = \sigma_0^{-2} (M\mathbf{x} - \mathbf{y})^T (M\mathbf{x} - \mathbf{y}), \tag{2}$$

$$J_2 = \mathbf{x}^T \mathrm{diag}\left(\boldsymbol{\sigma}_\mathbf{x}^{-2}\right)\mathbf{x}, \tag{3}$$

$$J_3 = \epsilon (D\mathbf{x})^T D\mathbf{x}, \tag{4}$$

where the term $J_1$ stands for the deviation of the model from the observation with scalar $\sigma_0$ to be a standard error of the observation; however, note that $\mathbf{y}$ and $M$ are prone to errors and cumulate uncertainties from measurement and the atmospheric transport model used for SRS calculations (including errors in the meteorological data used to drive the transport model); $J_1$ therefore includes errors in the model $M$, mapped into observation space; the term $J_2$ penalizes high values of the solution where the penalty is inverse proportional to the assumed standard errors of each source term element aggregated in vector $\boldsymbol{\sigma}_\mathbf{x}$, where symbol $\mathrm{diag}()$ denotes a diagonal matrix with an argument vector on its diagonal and zeros otherwise; and the term $J_3$ encourages smooth estimates of the source term $\mathbf{x}$ where $D$ is a tridiagonal differential matrix numerically approximating a Laplacian operator and the scalar $\epsilon$ weights the strength of the smoothness of the solution relative to the other two terms. Note that we assume the smoothness in time as it is used in (Stohl et al., 2011). This assumption may not be valid in cases such as explosions, which cause abrupt change in the source term.

Note that model (1) can be also used for problems with non-zero prior mean $\mathbf{x}_0$ and known covariance matrix of the observations, $R$. Using Choleski decomposition of the observation covariance matrix, $R = \Psi\Psi^T$, the original model can be written in form $\overline{\mathbf{y}} = \overline{M}\overline{\mathbf{x}} + \Psi\mathbf{e}$, where $\mathbf{e}$ is an isotropic noise and $\overline{\mathbf{x}}$ is assumed to have prior mean $\mathbf{x}_0$. Then, transformation $\mathbf{x} = \overline{\mathbf{x}} - \mathbf{x}_0$, $M = \Psi^{-1}\overline{M}$, $\mathbf{y} = \Psi^{-1}(\overline{\mathbf{y}} - \overline{M}\mathbf{x}_0)$ maps such a model into form (1) with zero mean prior and isotropic noise assumption.

This minimization problem leads to a system of linear equations that is solved for the source term $\mathbf{x}$. Since the solution is assumed to be positive, the optimization problem is subject to $\mathbf{x} \geq 0$:

$$\langle \mathbf{x} \rangle = \arg\min_x \left( J_1 + J_2 + J_3 \right), \quad \text{subject to } \mathbf{x} \geq 0. \tag{5}$$

This restrictions is achieved in the iterative algorithm by replacing all negative values by an arbitrary small positive number together with a reduction of their standard errors to force these values closer to the non-negative prior solution. This can be formalized by the selection of stop condition for ratio between negative and positive part of solution as $\frac{\langle \mathbf{x} \rangle_{neg}}{\langle \mathbf{x} \rangle_{pos}} < \delta$, where $\delta$ is a selected threshold. For the estimation algorithm, proper values of parameters $\sigma_0$, $\boldsymbol{\sigma}_\mathbf{x}$, and $\epsilon$ need to be preselected manually

**Algorithm 1** The main ideas of algorithm from (Eckhardt et al., 2008).

---

1. Iterate until sufficient solution $\langle \mathbf{x} \rangle$ is obtained:

    (a) Choose parameters $\sigma_0$, $\boldsymbol{\sigma}_{\mathbf{x}}$, $\epsilon$, and $\delta$.

    (b) Iterate until $\frac{\langle \mathbf{x} \rangle_{neg}}{\langle \mathbf{x} \rangle_{pos}} < \delta$ or maximum number of iteration is reached:

        i. Solve minimization problem given by Eq. (2)–(4).

        ii. Change negative parts of $\mathbf{x}$ to arbitrary small positive random numbers, reduce related variances $\boldsymbol{\sigma}_{\mathbf{x}}$ for negative parts of solution and increase variance $\boldsymbol{\sigma}_{\mathbf{x}}$ for positive parts of solution.

    (c) Report estimated source term $\langle \mathbf{x} \rangle$.

---

and potentially changed repeatedly until an acceptable solution is obtained. The main ideas of the algorithm are summarized in Algorithm 1.

## 2.2 Bayesian Interpretation of the Reference Method

The method of (Eckhardt et al., 2008) can be interpreted as a maximum *a posteriori* probability estimate of a particular probabilistic model. Specifically, the Gaussian observation model with truncated Gaussian prior distribution of the source term

$$p(\mathbf{y}|\mathbf{x}) = \mathcal{N}(M\mathbf{x}, \sigma_0^2 I_n) \propto \exp\left(-\frac{1}{2}\sigma_0^{-2}(M\mathbf{x} - \mathbf{y})^T(M\mathbf{x} - \mathbf{y})\right) \tag{6}$$

$$p(\mathbf{x}|\Sigma_{\mathbf{x}}) = t\mathcal{N}(\mathbf{0}, \Sigma_{\mathbf{x}}) \propto \exp\left(-\frac{1}{2}\mathbf{x}^T \Sigma_{\mathbf{x}} \mathbf{x}\right)\chi(x_i > 0), \tag{7}$$

$$\Sigma_{\mathbf{x}} = (\text{diag}\left(\boldsymbol{\sigma}_{\mathbf{x}}^{-2}\right) + \epsilon D^T D)^{-1}. \tag{8}$$

where $\mathcal{N}(\boldsymbol{\mu}, \Sigma)$ denotes a multivariate Gaussian distribution with mean $\boldsymbol{\mu}$ and covariance matrix $\Sigma$, $I_n$ is the $n \times n$ identity matrix, $t\mathcal{N}(\boldsymbol{\mu}, \Sigma, \langle 0, \infty \rangle)$ is a truncated Gaussian distribution with parameters $\boldsymbol{\mu}, \Sigma$ and support (domain) of physically realistic values restricted to positive values of all entries of the vector $\mathbf{x} = [x_1, \ldots, x_n]$, see Appendix A. The choice of Gaussian distribution is motivated primarily by tractability of its inference.

The logarithm of the posterior probability of the unknown $\mathbf{x}$ has the form

$$\log p(\mathbf{x}|\mathbf{y}) = -\frac{1}{2}(J_1 + J_2 + J_3) + \gamma(\mathbf{x}) + c, \tag{9}$$

where $\gamma(\mathbf{x})$ is the logarithm of the characteristic function enforcing positivity (see Appendix A), and $c$ aggregates all terms independent of $\mathbf{x}$. Maximization of the log-likelihood is then equivalent to minimization of the cost function of the reference method (5) where $\gamma(\mathbf{x})$ is the barrier function of the constraint on $\mathbf{x}$.

While interpretation of positivity by truncated normal distribution is non-standard, it has the same effect as the 'subject to' constraint. The maximum likelihood estimate is the value of $\mu$ if $\mu > 0$ and it is zero otherwise, Fig. 1.

The maximum likelihood solution is the simplest case of Bayesian inference. Application of full Bayesian inference (i.e. evaluation of full posterior distribution and their marginals) can address two important problems:

1. selection of tuning parameters $\sigma_0$, $\boldsymbol{\sigma_x}$, and $\epsilon$ which are considered to be hyper-parameters and estimated from the data,

2. selection of the appropriate model $M$ via Bayesian model selection.

We are concerned only with the first problem in this paper while matrix $M$ is considered fixed. Extension of the proposed methodology to Bayesian model selection is possible (Bishop, 2006), however it is rather long and its proper treatment is beyond the scope of this paper. Full Bayesian treatment of the unknowns $\sigma_0$, $\boldsymbol{\sigma_x}$, and $\epsilon$ is not analytically tractable. Approximate inference of $\sigma_0$ and $\boldsymbol{\sigma_x}$ is possible, however estimation of $\epsilon$ in represents a challenge since the determinant of the covariance becomes too complex. Moreover, common variance of temporal derivative of the source term may not be realistic, since it is subject to abrupt changes cause e.g. by explosions. Therefore, we will present results for a different and more complex structure of the prior variance $\Sigma_{\mathbf{x}}$ that allows stable and reliable estimation of the source term vector $\mathbf{x}$ via the Variational Bayes method.

## 3    Probabilistic Model with Unknown Prior Covariance

We formulate the probabilistic model to cope with the linear inverse problem, Eq. (1), and derive an iterative algorithm to estimate parameters of this model.

### 3.1    Observation Model

The observation model is identical to Eq. (6), i.e. the isotropic Gaussian noise model[1]. However, we will consider the precision (inverse variance) of the observations to be unknown, parameterized by $\omega^{-1} = \sigma_0^2$,

$$p(\mathbf{y}|\mathbf{x},\omega) = \mathcal{N}_{\mathbf{y}}\left(M\mathbf{x},\omega^{-1}I_p\right). \tag{10}$$

Since $\omega$ is unknown and will be estimated, similarly to Tipping and Bishop (1999), we define its own prior distribution in the form of the Gamma distribution (which is conjugate prior for precision of the Gaussian distribution) as

$$p(\omega) = \mathcal{G}\left(\vartheta_0,\rho_0\right), \tag{11}$$

with chosen prior constants $\vartheta_0, \rho_0$. These constants are needed for numerical stability, however, they are set as low as possible to provide non-informative prior, see Algorithm 2.

Note that the model with different elements on diagonal of the covariance matrix of measurements were also studied in literature, see e.g. Michalak et al. (2005). Modification of the proposed algorithm to diagonal precision matrix with unknown elements on the main diagonal is very simple. However, such a model was found to be susceptible to local optima than the presented model. The presented model was found to be more reliable in practical tests.

### 3.2    Prior Model

We use the same prior for the source term vector as in Eq. (7), with the exception that the prior covariance of $\mathbf{x}$, denoted as $\Sigma_{\mathbf{x}}$, is unknown. Note from Eq. (8) that the covariance matrix is a band matrix with predefined structure; tridiagonal matrix in this

---

[1]Gaussian noise with an arbitrary known covariance matrix can be transformed into this form by scaling of the observations and the SRS matrix.

case. Relaxing the assumption of the tridiagonal structure we consider the following structure of the prior covariance:

$$\Sigma_\mathbf{x} = L \Upsilon L^T. \tag{12}$$

It is composed of diagonal matrix $\Upsilon = \mathrm{diag}(\boldsymbol{v})$, with unknown positive diagonal entries forming vector $\boldsymbol{v} = [v_1, \ldots, v_n]$ and zeros otherwise. $L$ is a lower bidiagonal matrix

$$5 \quad L = \begin{pmatrix} 1 & 0 & 0 & 0 \\ l_1 & 1 & 0 & 0 \\ 0 & \ddots & 1 & 0 \\ 0 & 0 & l_{n-1} & 1 \end{pmatrix}, \tag{13}$$

with unknown off-diagonal elements forming a vector $\mathbf{l} = [l_1, \ldots l_{n-1}]$. This considerations preserve the tridiagonal structure of the covariance matrix $\Sigma_\mathbf{x}$ and allow us to model each diagonal separately. The task is to introduce prior models for vectors $\boldsymbol{v}$ and $\mathbf{l}$ whose estimates fully determine the decomposition in Eq. (12).

The prior model of the vector $\boldsymbol{v}$ is selected as

$$10 \quad p(v_j) = \mathcal{G}_{v_j}(\alpha_0, \beta_0), \quad \forall j = 1, \ldots, n, \tag{14}$$

where $\alpha_0, \beta_0$ are selected non-informative prior constants, see Algorithm 2. The prior model of the vector $\mathbf{l}$ is selected in a problem specific way. Note that for $l_j = 0$, model in Eq. (12) corresponds to Eq. (8) with $\epsilon = 0$. For $l_j = -1$, model in Eq. (12) corresponds to Eq. (8) with $\epsilon \to \infty$. Since we expect the result to be within this interval, we define the prior on $l_j$ to be independently Gaussian distributed with unknown precision $\psi_j$:

$$15 \quad p(l_j | \psi_j) = \mathcal{N}_{l_j}(l_0, \psi_j^{-1}), \tag{15}$$

$$p(\psi_j) = \mathcal{G}_{\psi_j}(\zeta_0, \eta_0), \quad \forall j = 1, \ldots, n-1, \tag{16}$$

where $\zeta_0, \eta_0$ are selected prior constants. Since we expect the neighboring values $x_i$ and $x_{i+1}$ to be either uncorrelated ($l_j = 0$) or correlated ($l_j = -1$) we choose parameters $l_0, \zeta_0, \eta_0$ to cover these extremes with preference for a value $l_0 = -1$, and precision $\psi_j$ set around this value using selection $\zeta_0 = \eta_0 = 10^{-2}$. This allows parameter $l_j$ to vary in the range circa $-1 \pm 100$
20 which we consider to be sufficiently non-informative. Lower values of $\zeta_0$ and $\eta_0$ results in posterior estimates closer to $l_0$. On the other hand, further relaxation of these parameters to a wider range results in higher sensitivity to local extremes and potentially numerical instability.

The joint model of the full distribution is then:

$$25 \quad p(\mathbf{y}, \mathbf{x}, \boldsymbol{v}, \mathbf{l}, \psi_{1,\ldots,n-1}, \omega) = p(\mathbf{y}|\mathbf{x}, \omega) p(\mathbf{x}|\boldsymbol{v}, \mathbf{l}, \psi_{1,\ldots,n-1}) p(v_n) \prod_{i=1}^{n-1} p(v_i) p(l_i|\psi_i) p(\psi_i). \tag{17}$$

Estimation of all unknown parameters can be obtained by the Bayes rule which is however computationally intractable.

## 3.3 Iterative Variational Bayes Algorithm

Following the Variational Bayesian methodology (Šmídl and Quinn, 2006), we seek a posterior distribution in a very specific form, satisfying posterior conditional independence:

$$p(\mathbf{x}, \boldsymbol{v}, \mathbf{l}, \psi_{1,\ldots,n-1}, \omega | \mathbf{y}) \approx p(\mathbf{x}|\mathbf{y})p(\boldsymbol{v}|\mathbf{y})p(\mathbf{l}|\mathbf{y})p(\psi_{1,\ldots,n-1}|\mathbf{y})p(\omega|\mathbf{y}). \tag{18}$$

5     The best possible approximation is defined as a minimizer of the Kullback-Leibler divergence (Kullback and Leibler, 1951) between the solution and the hypothetical true posterior. The choice of this form is motivated by simplicity of evaluation and experience indicates that it is a very good approximation for linear models (Bishop, 2006; Šmídl and Quinn, 2006).

The necessary conditions of the best approximation uniquely determine the form of the posterior distributions. These were identified to be as follows:

$$\tilde{p}(\mathbf{x}|\mathbf{y}) = t\mathcal{N}_{\mathbf{x}}\left(\mu_{\mathbf{x}}, \Sigma_{\mathbf{x}}\right), \tag{19}$$

$$\tilde{p}(v_j|\mathbf{y}) = \mathcal{G}_{v_j}\left(\alpha_j, \beta_j\right), \quad \forall j = 1, \ldots, n, \tag{20}$$

$$\tilde{p}(l_j|\mathbf{y}) = \mathcal{N}_{l_j}\left(\mu_{l_j}, \Sigma_{l_j}\right), \quad \forall j = 1, \ldots, n-1, \tag{21}$$

$$\tilde{p}(\psi_j|\mathbf{y}) = \mathcal{G}_{\psi_j}\left(\zeta_j, \eta_j\right), \quad \forall j = 1, \ldots, n-1, \tag{22}$$

$$\tilde{p}(\omega|\mathbf{y}) = \mathcal{G}_{\omega}\left(\vartheta, \rho\right). \tag{23}$$

The shaping parameters of posterior distributions, Eq. (19)–(23), $\mu_{\mathbf{x}}, \Sigma_{\mathbf{x}}, \alpha_j, \beta_j, \mu_{l_j}, \Sigma_{l_j}, \zeta_j, \eta_j, \vartheta, \rho$, derived according to the standard Variational Bayes procedure, see (Šmídl and Quinn, 2006), are given in Appendix B. The shaping parameters are functions of standard moments of posterior distributions, e.g. $\langle\mathbf{x}\rangle$, $\langle\mathbf{x}\mathbf{x}^T\rangle$ and $\langle\mathbf{x}^T\mathbf{x}\rangle$ for the distribution $\tilde{p}(\mathbf{x}|\mathbf{y})$. Symbol $\langle\mathbf{x}\rangle$ denotes the expected value with respect to the distribution on the variable in the argument. The shaping parameters and the required moments form a set of implicit equations which is solved iteratively using Algorithm 2. Good initialization

should be considered since convergence only to the local minima is guaranteed (Šmídl and Quinn, 2006). We propose to initialize the algorithm by solution of the ordinary least squares with Tikhonov regularization tuned such that the data and the regularization term have equal scale. It is achieved by choice of the initial value of the estimate of the precision parameter $\langle\omega\rangle^{(0)} = \frac{1}{\max(M^T M)}$. Here, superscript $^{(i)}$ is used to denote iteration number of the algorithm. The algorithm will be called Least Squares with Adaptive Prior Covariance (LS-APC) and is freely available for download from http://www.utia.cz/linear_

inversion_methods.

Note that the algorithm is closely related to Algorithm 1 of the reference method. It also iteratively solves the least squares problem but with adaptive tuning of its parameters. The proposed method has the following differences:

1. The algorithm is largely tuning-free, i.e. all hyper-parameters $\omega, l_i, v_i, \psi_i$ are estimated from the data. The results may slightly differ for different choices of the initial conditions since the Variational Bayes solution may suffer from local

minima. The most sensitive initial value is $\langle\Upsilon\rangle^{(0)}$ of tuning parameter $\gamma$. The sensitivity of the solution to this initial choice is very low which will be discussed in Sect. 5.2.

**Algorithm 2** Least Square with Adaptive Prior Covariance (LS-APC) algorithm.

---

1. Initialization

   (a) Set prior parameters $\vartheta_0, \rho_0, \alpha_0, \beta_0$ to non-informative values of $10^{-10}$ (yielding non-informative priors). Values $\zeta_0, \eta_0$ are set to physically meaningful values of $10^{-2}$.

   (b) Set initial values (denoted by zero iteration number in superscript $^{(0)}$) used in computation of the covariance matrix of the source term, $\Sigma_{\mathbf{x}}$: $\langle \omega \rangle^{(0)} = \frac{1}{\max(M^T M)}$, $\langle \Upsilon \rangle^{(0)} = \gamma I_n$, and $\langle L \rangle^{(0)} = I_n$. If $\gamma$ is not specified use $\gamma = 1$.

   (c) Set iteration index $i = 1$.

2. Iterate until convergence or maximum number of iterations is reached:

   (a) Compute estimate of the source term $\langle \mathbf{x} \rangle^{(i)}$ using least squares:

   $$\Sigma_{\mathbf{x}}^{(i)} = \left( \langle \omega \rangle^{(i-1)} M^T M + \left\langle L \Upsilon L^T \right\rangle^{(i-1)} \right)^{-1}, \tag{24}$$

   $$\mu_{\mathbf{x}}^{(i)} = \Sigma_{\mathbf{x}}^{(i)} \left( \langle \omega \rangle^{(i-1)} M^T \mathbf{y} \right), \tag{25}$$

   using moments of the truncated normal distribution, Eq. (A).

   (b) Update estimates of $\langle \Upsilon \rangle^{(i)}$ and $\langle L \rangle^{(i)}$, using Eq. (B2)–(B4) defined in Appendix B,

   (c) Compute precision parameter $\langle \omega \rangle^{(i)}$ using Eq. (B5) in Appendix B.

3. Report estimated source term $\langle \mathbf{x} \rangle$ and its uncertainty $\Sigma_{\mathbf{x}}$.

---

2. Since estimating the hyper-parameter values requires the calculation of the variance of the posterior distribution, the covariance matrix of the least squares problem needs to be evaluated; the cost of this operation is circa $O(n^{2.4})$ in each iteration. This implies a slightly higher computational cost compared to Algorithm 1 where this matrix is not needed.

3. The method of positivity enforcement is replaced by moments of the truncated normal distribution.

## 4    Verification Using a Synthetic Dataset

To test the proposed LS-APC algorithm and to demonstrate its performance, we design a synthetic dataset before performing a real data experiment. We generate elements of the matrix $M \in \mathbf{R}^{20 \times 10}$ as random samples from an uniform distribution between 0 and 1 and elements less then 0.5 were cropped to 0 to reduce the condition number of the matrix $M$ (which is 6.69 in l2-norm in this dataset). The source term is generated as $\mathbf{x}_{\text{true}} = [0, 0, 0, 1, 1, 1, 0, 0, 0, 0]$ as shown in Fig. 2, top row, using dashed red line. The vector of measurement data is generated according to the assumed model in Eq. (1) as $\mathbf{y} = M\mathbf{x} + \mathbf{e}$ where three sets are generated with the same matrix $M$ but with different levels of the noise term $\mathbf{e}$. Each element $e_j$ is generated randomly as $\mathcal{N}(0, c_k^2)$ where the coefficients are set as $c_1 = 0$ for the set 1, $c_2 = 0.4$ for the set 2, and $c_3 = 0.8$ for the set 3.

Then, negative elements of $\mathbf{y}$ are cropped to 0. Note that these data are supplied together with the LS-APC algorithm as a tutorial example.

The results from the LS-APC algorithm for this dataset are given in Fig. 2. The estimates of the source term are shown in the uppermost row (solid blue lines) together with the simulated true source term (dashed red line). The estimated values of the vector $\langle \boldsymbol{\upsilon} \rangle$, i.e. the diagonal of the matrix $\langle \Upsilon \rangle$, are displayed in the second row. This parameter models the sparsity of the solution where a higher value signifies higher confidence that the corresponding element of the solution is zero. The parameter $\langle \mathbf{l} \rangle$ modeling the smoothness of the solution is shown in Fig. 2, bottom row. Note that at the constant parts of the solution this parameter is approaching -1, signifying highly correlated neighboring elements, while it is approaching zero at the time of the step change, indicating uncorrelated neighbors. The two parameters $\langle \boldsymbol{\upsilon} \rangle$ and $\langle \mathbf{l} \rangle$ can also compensate one another, as is demonstrated on the falling edge of the source term. Instead of the expected zero in the smoothness parameter $l_6$, the posterior value is close to the prior. The difference in the data is compensated by the sparsity parameter $\upsilon_6$ which is very low, indicating very low confidence in this value.

The quality of the reconstruction depends on the noise level, as demonstrated in individual columns of Fig. 2. As expected, the source term is reconstructed precisely when the data are noise free (Fig. 2, left column). With increasing noise, the reconstruction departs form the ground truth (Fig. 2, middle column), however, the start and end of the release is still estimated with sharp raising and falling edges. The estimate is also sparse, i.e. the estimated values of the source term outside of the true release window are zero. Note that this result was achieved with standard deviation of the noise equal to $40\%$ of the released quantity. Naturally, with even higher noise (standard deviation equal to 80% of the released quantity, Fig. 2, right column), the estimates also depart from the ideal shape and yields undesired artifacts.

## 5 Experimental Results for the ETEX Data

The European Tracer Experiment (ETEX) took place at Monterfil in Brittany, France, on 23th October 1994 (Nodop et al., 1998). Its attractiveness is that it is one of a very few controlled large-scale tracer release experiments with a large amount of available information, see https://rem.jrc.ec.europa.eu/etex/. During ETEX, two release experiments were made. We use here only the data from the first experiment (ETEX-I), for which atmospheric dispersion models generally performed much better than for the second experiment, e.g., (Stohl et al., 1998). During ETEX-I, a total amount of 340 kilograms of perfluoromethyl-cyclohexane (PMCH) was released at a constant rate during nearly 12 hours. PMCH is nearly inert in the atmosphere and does not experience dry deposition or wet scavenging and is thus suitable for testing how well transport models can handle atmospheric dispersion. Atmospheric concentrations of the released PMCH were monitored across Europe by a network of 168 measurement stations with a sampling interval of 3 hours over a period of 72 hours. The release location and station locations are shown in Fig. 3. The ETEX data set has been used previously for testing inverse models, e.g., by (Bocquet, 2005a, 2007), (Krysta et al., 2008), and (Martinez-Camara et al., 2014).

To construct the SRS matrix $M$, we used version 8.1 of the Lagrangian particle dispersion model FLEXPART (Stohl et al., 2005, 1998). An earlier version of the model was evaluated against the first ETEX experiment and revealed relatively good

performance compared to other models (Stohl et al., 1998). We assumed that the release location is known, and that the release occurred during a 5-day period including the true release time but that the source term (i.e., released amount as a function of time) is not known. Thus, we performed 120 forward calculations from the release site, each for a hypothetical unit release during a one-hour period. For each of these unit release simulations, the simulated tracer concentrations were sampled at all the measurement station locations and during the exact measurement times (in total, 3102 measurements were made), to construct the SRS matrix M of the size $3102 \times 120$. The SRS matrix was used together with the observation vector, $\mathbf{y}$, of size $3102 \times 1$, to reconstruct the source term, vector $\mathbf{x}$ of the size $120 \times 1$. The reconstructed source term can then be compared with the known true source term, to evaluate the skill of the reconstruction. The same set-up was used by Martinez-Camara et al. (2014) to test a method to blindly remove outliers.

For running FLEXPART, we have used meteorological input data from the European Center for Medium-Range Weather Forecasts (ECMWF). Different data sets are available from ECMWF and we have used two different data sets: 1) Data from the 40-year re-analysis (ERA-40); 2) Data from the continuously updated ERA-Interim re-analysis. For both meteorological data sets we have run FLEXPART in two different configurations:

A)     with the model time step in the boundary layer limited to less than 20% of the Lagrangian timescale and a maximum value of 300 s;

B)     with time step only limited by 300 s, which may be chosen for computationally demanding real-time simulations.

The Lagrangian time scale depends on the turbulence conditions in the planetary boundary layer and is computed in FLEXPART for every particle at every individual time step. Lagrangian time scales can be very short (order of seconds) and, thus option A requires very short numerical integration time steps. Close to a source, this is the only accurate way of ensuring the well-mixed condition and a correct simulation of near-field dispersion. Over longer transport distances, such an accurate description of small-scale turbulent transport is often not necessary as transport errors are dominated by other sources of error (such as errors in large-scale wind fields). Thus, compromises are often made in numerical simulations, especially for real-time model applications, where longer time steps are used. This is explored with configuration B.

While the differences between these simulations are actually rather small in terms of simulated SRS values, they can serve as a lower estimate of the uncertainty associated with the SRS calculations and can still produce quite substantial differences in the retrieved source terms.

## 5.1   Source Term Estimation Using LS-APC Algorithm

The task is to estimate the original source term $\mathbf{x}$ based only on the available measurement data. The Algorithm 2 was applied to the selected example data ETEX ERA-Interim B and the results are presented in Fig. 4. In the top panel of Fig. 4, the red line denotes the true source term while the blue line denotes the estimated source term $\langle \mathbf{x} \rangle$ accompanied by the 99% highest posterior density region is given using gray fill region. The estimated sparsity parameter $\langle \boldsymbol{v} \rangle$, i.e. the diagonal of the matrix $\langle \Upsilon \rangle$, is given in the middle panel of Fig. 4, and the estimated smoothness parameter $\langle \mathbf{l} \rangle$, i.e. second diagonal of the matrix $\langle L \rangle$, is given in the bottom panel. Note that the sparsity parameter is approaching $10^{10}$ (value determined by $\alpha_0$ and $\beta_0$) at times

when no release occurred; therefore, the posterior mean value is close to the prior value, which is zero. The posterior mean value of the smoothness parameter $l_j$ is $-1$ when the neighboring values of the solution are close to each other. During periods of rapid change of the release, the estimate of the smoothness parameter approaches zero.

While the reconstructed source term does not agree exactly with the known source profile, the true total of the source term, i.e. 340kg, is on the edge of the 99% highest posterior density region which is (120,340)kg. This result is achieved without any tuning of the internal parameters of the FLEXPART dispersion model. Also the timing of the release is well captured, although the reconstructed release shows some variation during the release period, while the true release rate was constant. Furthermore, the reconstruction suggests some small release to occur also after the true release has ended. The quality of the reconstruction is comparable to or better than previous reconstructions of the ETEX source term (e.g., Seibert and Stohl, 1999; Bocquet, 2005a, 2007; Martinez-Camara et al., 2014. Note that these results were obtained without setting any tuning parameters, all regularization parameters are estimated from the data within the iterative algorithm. The sensitivity of this approach to the initial values and assumed uncertainties will be studied in comparison with other algorithms.

## 5.2 Comparison and Sensitivity Study

We compare results from the proposed LS-APC algorithm, Algorithm 2, with those obtained from (Eckhardt et al., 2008), Algorithm 1, with the RegClean algorithm (Martinez-Camara et al., 2014), the least absolute shrinkage and selection operator (LASSO) (Tibshirani, 1996), and the Tikhonov regularization (Golub et al., 1999). Specifically, we study the ability of the proposed solution to regularize the problem for different choices of the selected tuning parameters. It was found that the results of Algorithms 1 and 2 are most sensitive to initial values of the sparsity parameters, $\boldsymbol{\sigma_x}$ and $\boldsymbol{\upsilon}$ respectively. Similarly, the RegClean, LASSO, and Tikhonov algorithms also have parameters influencing preference for penalization of large values of the solution (e.g. the $\alpha$ parameter of the Tikhonov and LASSO regularization). Thus, we run all five algorithms with this selected tuning parameter set in points of interval $\alpha \in < e^{-15}, e^{+7} >$ for four ETEX datasets. That is, for the methods with diagonal choice, e.g. the reference method in Algorithm 1, we set $\boldsymbol{\sigma}_x^{-2} = \alpha I_n$. For the LS-APC algorithm, this choice influences only the initial value of the regularization parameter via $\gamma = \alpha$.

All remaining parameters of the other methods were kept at their default values (RegClean) or set to best performing values (Algorithm 1). Evaluation of the results was performed on the metric of mean absolute error (MAE) between the true and the estimated source term:

$$\text{MAE} = \frac{1}{n} \sum_{j=1}^{n} |x_{j,\text{true}} - x_{j,\text{estim}}|. \tag{26}$$

The computed MAEs between the true source term and the estimated source term for all methods and for the same range of the tuning parameter $\alpha$ are displayed in Fig. 5 for ETEX ERA-40, and in Fig. 6 for ETEX ERA-Interim, top rows. The estimates of the total released mass for all methods and for the same range of the tunning parameter $\alpha$ are displayed in the bottom row of Fig. 5 and 6. Note that all methods achieve similar results although for different values of the tuning parameters. This is most obvious in the estimate of the total released mass, where each method has a range of tuning values yielding the same estimate. This looks like a plateau on the curve. The value of the total released mass at this plateau is very similar for all

methods. The exception is the experiment ERA-Interim A, where the curves of the estimated total released mass contain two plateaus. Comparison with the true total released mass of 340 kg is misleading in this case since the plateau at 180 kg is due to an artifact, as discussed below.

Examples of results from all algorithms for all settings of the regularization parameter for ETEX ERA-Interim B are displayed in Fig. 7 and for the problematic mode of ETEX ERA-Interim A in Fig. 8. The true source term is denoted by the dashed red lines and the estimated source terms are denoted using blue lines. For LS-APC, all estimates are overlapping, for algorithms sensitive to this choice, the lines form an area. Note that the ETEX ERA-Interim A has a strong artifact at the first element of the solution since all receptors have high sensitivity to it (high values in the first column of the SRS matrix). Thus, non-zero value of the first element of $\mathbf{x}$ can explain a part of the observation.

Note that the LS-APC algorithm provides results that are almost insensitive to the value of the tuning parameter (used only as a starting point). Moreover, the results of the LS-APC algorithm correspond to the results of other methods with best tuned parameters. However, the proposed algorithm still suffers form local minima as demonstrated in the case of ETEX ERA-Interim A. However, the same local minima are visible for the other methods as well. Despite this non-uniqueness, the algorithm still provides only two possible solutions in contrast to the other algorithms that yield a range of possible solutions for different settings of the tuning parameters as can be seen in Fig. 7 and 8.

The computational cost of the proposed algorithm is higher than simple techniques such as LASSO and the Tikhonov regularization since least squares fit calculations are run in each iteration. The convergence is typically reached in tens of iterations. All experiments were run with 100 iterations where the equilibrium was always reached. The runtime of the full iterative algorithm is 3.3 seconds for $n = 120$ on a conventional PC with Intel Core i7-870 CPU. Scaling of the algorithm to higher dimension is dominated by the inverse of $n \times n$ matrix which scales with $O(n^{2.34})$.

## 6  Conclusions

We present a novel algorithm for the linear inversion problem which is applied to the problem of source term determination for pollutant releases into the atmosphere. It is closely related to the common optimization based techniques with regularization. The model is based on a probabilistic formulation with unknown prior covariance matrix. Application of the Variational Bayes method to the proposed probabilistic model results in an iterative algorithm that is closely related to the existing algorithms. The key difference is that the new algorithm estimated all hyper-parameters from the data without human interaction.

The proposed algorithm was validated using data from the ETEX experiment. It was shown that the LS-APC algorithm provides more consistent estimates of the source term with very little influence from initialization and with no need of human interaction. Therefore, the algorithm seems particularly suited for real-time applications where there is no time for manually setting tuning parameters.

## Code availability

The code for the LS-APC algorithm is available upon request for academic and non-commercial use from the corresponding author or through the following link: http://www.utia.cz/linear_inversion_methods. The implementation is provided in MATLAB while no additional toolboxes are required for the algorithm.

*Acknowledgements.* This research is supported by EEA/Norwegian Financial Mechanism under project MSMT-28477/2014 Source-Term Determination of Radionuclide Releases by Inverse Atmospheric Dispersion Modelling (STRADI).

## Appendix A:  Truncated Normal Distribution

Truncated normal distribution, denoted as $t\mathcal{N}$, of a scalar variable $x$ on interval $[a;b]$ is defined as

$$t\mathcal{N}_x(\mu,\sigma,[a,b]) = \frac{\sqrt{2}\exp((x-\mu)^2)}{\sqrt{\pi}\sigma(erf(\beta)-erf(\alpha))}\chi_{[a,b]}(x), \tag{A1}$$

where $\alpha = \frac{a-\mu}{\sqrt{2}\sigma}$, $\beta = \frac{b-\mu}{\sqrt{2}\sigma}$, function $\chi_{[a,b]}(x)$ is a characteristic function of interval $[a,b]$ defined as $\chi_{[a,b]}(x) = 1$ if $x \in [a,b]$ and $\chi_{[a,b]}(x) = 0$ otherwise. $\mathrm{erf}()$ is the error function defined as $\mathrm{erf}(t) = \frac{2}{\sqrt{\pi}}\int_0^t e^{-u^2}\,\mathrm{d}u$.

The moments of truncated normal distribution are

$$\langle x\rangle = \mu - \sqrt{\sigma}\frac{\sqrt{2}[\exp(-\beta^2)-\exp(-\alpha^2)]}{\sqrt{\pi}(\mathrm{erf}(\beta)-\mathrm{erf}(\alpha))}, \tag{A2}$$

$$\langle x^2\rangle = \sigma + \mu\widehat{x} - \sqrt{\sigma}\frac{\sqrt{2}[b\exp(-\beta^2)-a\exp(-\alpha^2)]}{\sqrt{\pi}(\mathrm{erf}(\beta)-\mathrm{erf}(\alpha))}. \tag{A3}$$

For multivariate case, see (Šmídl and Tichý, 2013).

## Appendix B:  Shaping Parameters of Posterior Distributions

$$\Sigma_{\mathbf{x}} = \left(\langle\omega\rangle M^T M + \langle L\Upsilon L^T\rangle\right)^{-1}, \qquad\qquad \mu_{\mathbf{x}} = \Sigma_{\mathbf{x}}\left(\langle\omega\rangle M^T\mathbf{y}\right), \tag{B1}$$

$$\alpha = \alpha_0 + \frac{1}{2}\mathbf{1}_{n,1}, \qquad\qquad \beta = \beta_0 + \frac{1}{2}\mathrm{diag}\left(\langle L^T\mathbf{x}\mathbf{x}^T L\rangle\right), \tag{B2}$$

$$\Sigma_{l_j} = \left(\langle v_j\rangle\langle x_{j+1}^2\rangle + \langle\psi_j\rangle\right)^{-1}, \qquad\qquad \mu_{l_j} = \Sigma_{l_j}\left(-\langle v_j\rangle\langle x_j x_{j+1}\rangle + l_0\langle\psi_j\rangle\right), \tag{B3}$$

$$\zeta_j = \zeta_0 + \frac{1}{2}, \qquad\qquad \eta_j = \eta_0 + \frac{1}{2}\langle(l_j-l_0)^2\rangle, \tag{B4}$$

$$\vartheta = \vartheta_0 + \frac{p}{2}, \qquad\qquad \rho = \rho_0 + \frac{1}{2}\mathrm{tr}\left(\langle\mathbf{x}\mathbf{x}^T\rangle M^T M\right) - \frac{1}{2}2\mathbf{y}^T M\langle\mathbf{x}\rangle + \frac{1}{2}\mathbf{y}^T\mathbf{y}. \tag{B5}$$

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

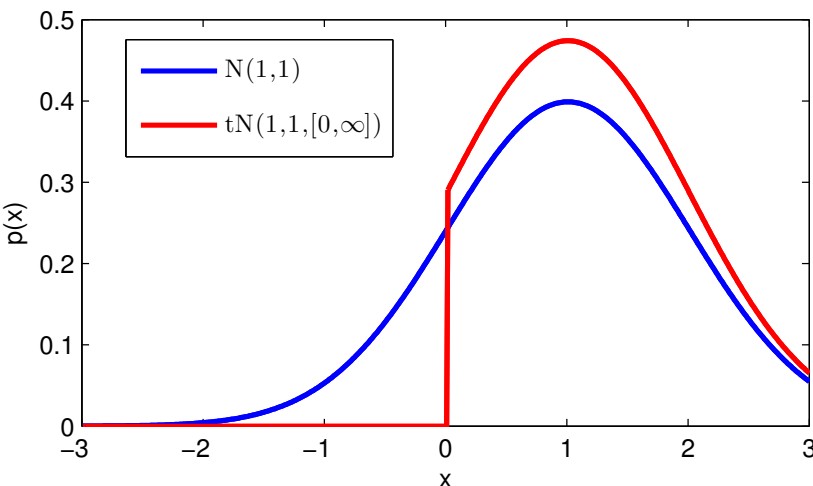

**Figure 1.** Example of the normal distribution $\mathcal{N}(1,1)$, blue line, and the truncated normal distribution $t\mathcal{N}(1,1,[0,\infty])$, red line.

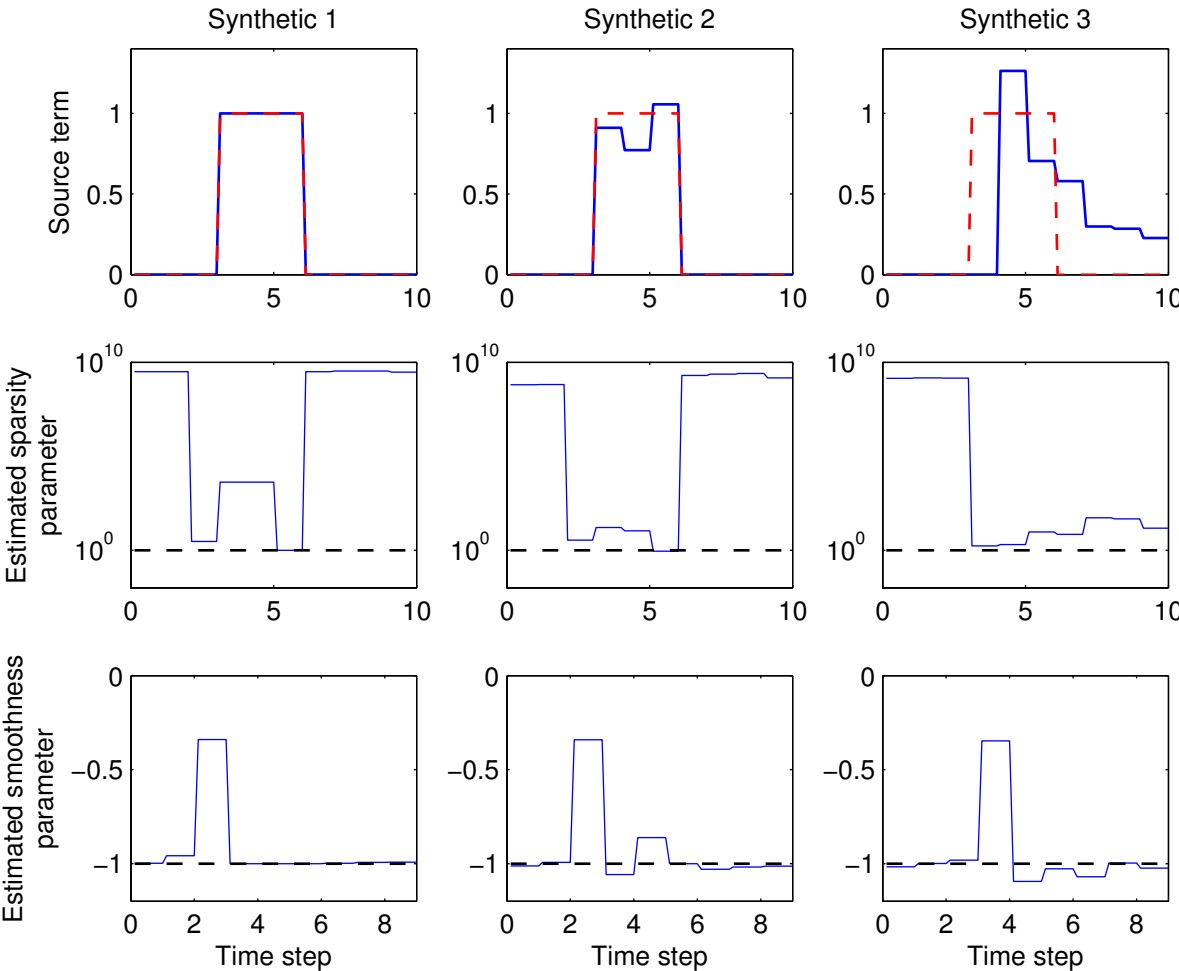

**Figure 2.** The results of the LS-APC algorithm on synthetically generated dataset with different levels of noise degradation (increasing from left to right; $e_j = \mathcal{N}(0, c_k^2)$, where $c_k = 0$ for the set Synthetic 1, $c_k = 0.4$ for the set Synthetic 2, and $c_k = 0.8$ for the set Synthetic 3). In the top panel, the true source term is given by the red line while the estimated source term is given by the blue line. The estimated sparsity parameters, vectors $\langle \upsilon \rangle$, are given in the middle panel using full line while prior values are given using dashed black lines and the estimated smoothness parameters, vectors $\langle l \rangle$, are given in the bottom panel while prior values are given using dashed black lines.

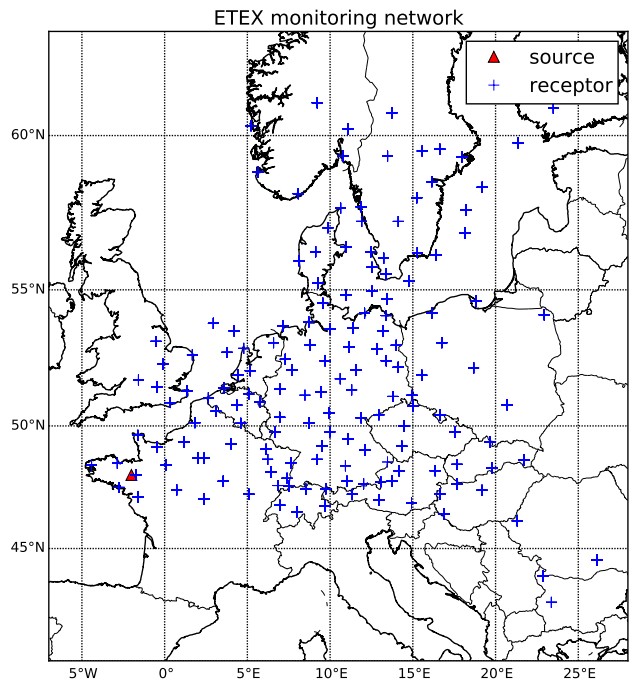

**Figure 3.** Domain of the ETEX experiment with source (red triangle) and receptors (blue crosses).

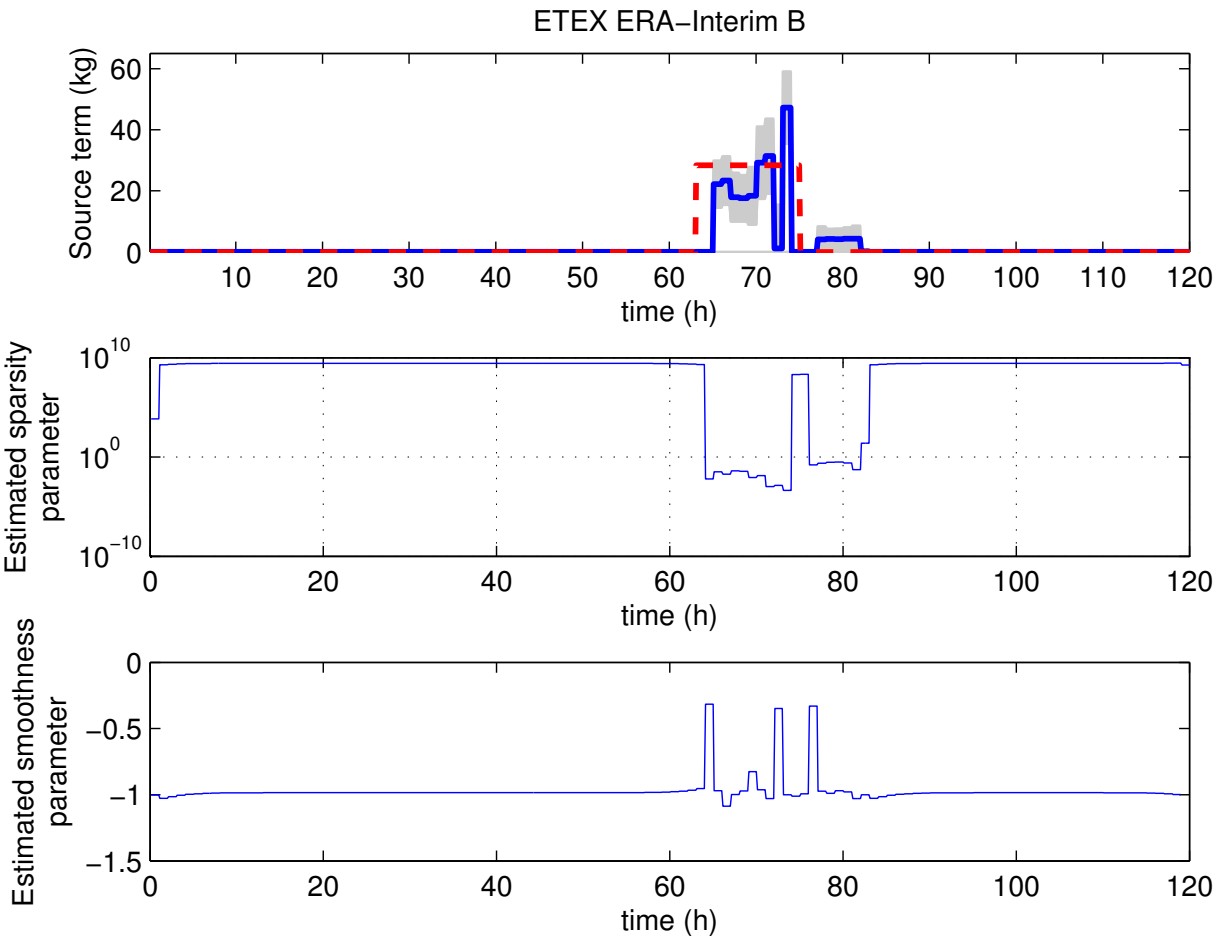

**Figure 4.** The results of the LS-APC algorithm for the ETEX experiment (ETEX ERA-Interim B). In the top panel, the true source term is given by the red line while the estimated source term is given by the blue line associated with the 99% highest posterior density region using gray filled regions. The estimated sparsity parameter, vector $\langle \boldsymbol{v} \rangle$, is given in the middle panel and the estimated smoothness parameter, vector $\langle \mathbf{l} \rangle$, is given in the bottom panel.

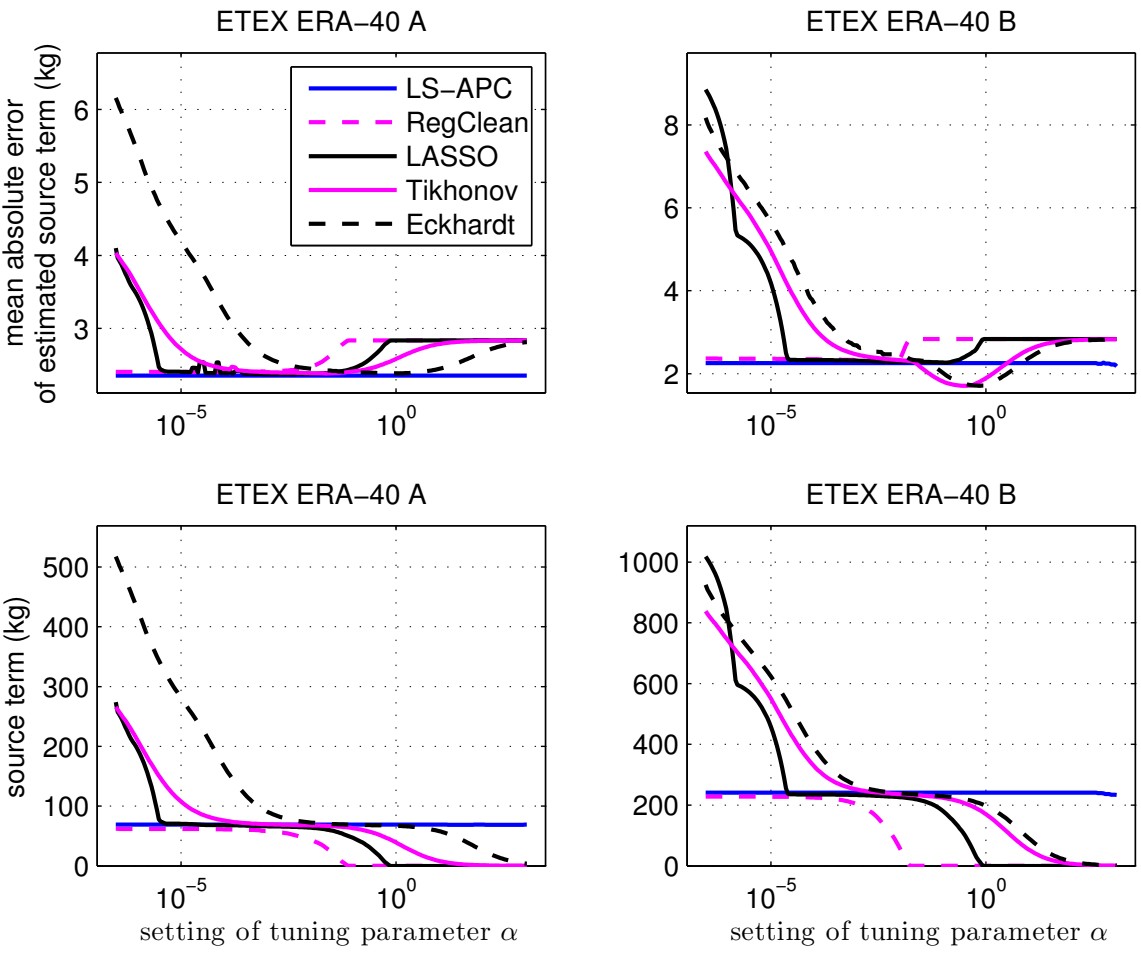

**Figure 5.** Comparison of sensitivity of the tested algorithms to the setting of the selected tuning parameter $\alpha$ measured in terms of the mean absolute error metric (top row), Eq. (26), and total estimated mass of the source term (bottom row) on data ETEX ERA-40 A and B.

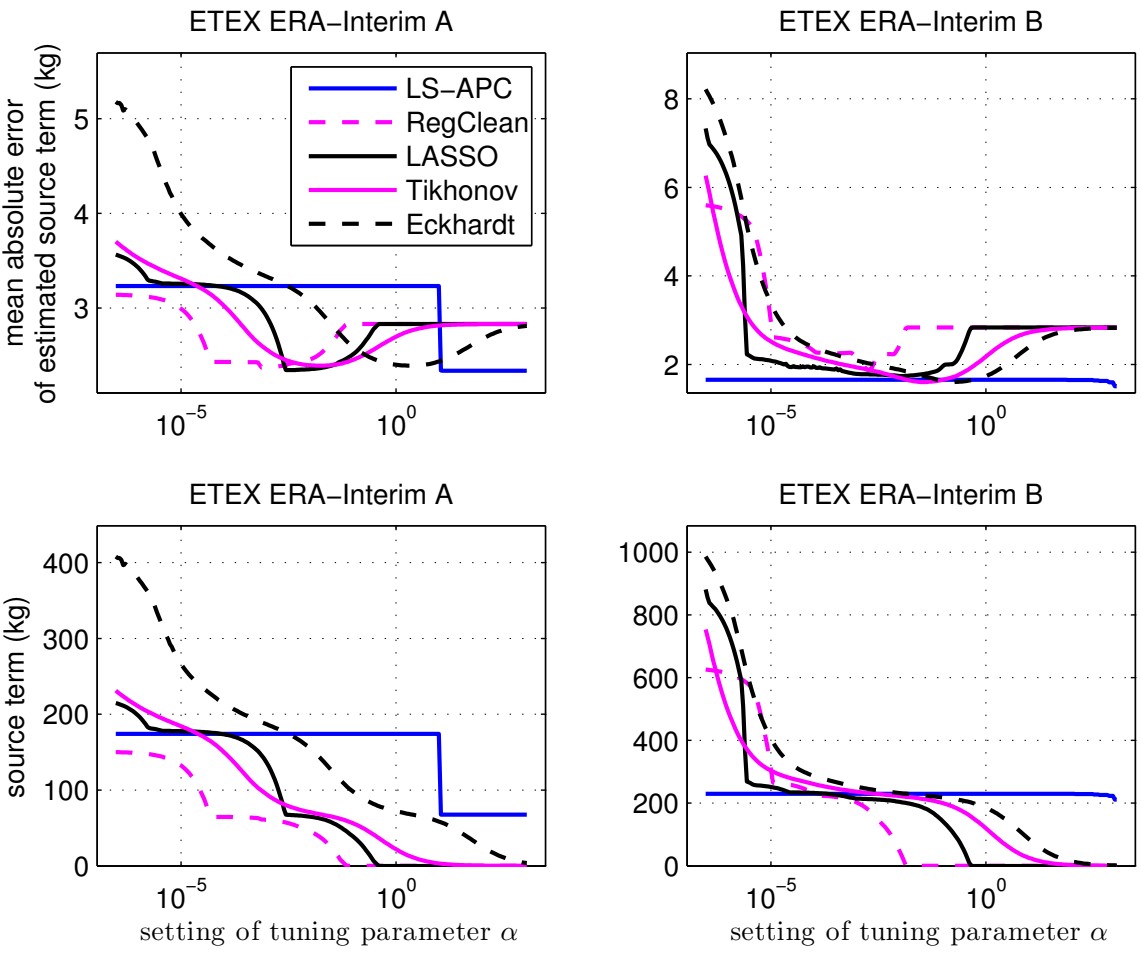

**Figure 6.** Comparison of sensitivity of the tested algorithms to the setting of the selected tuning parameter $\alpha$ measured in terms of the mean absolute error metric (top row), Eq. (26), and total estimated mass of the source term (bottom row) on data ETEX ERA-Interim A and B.

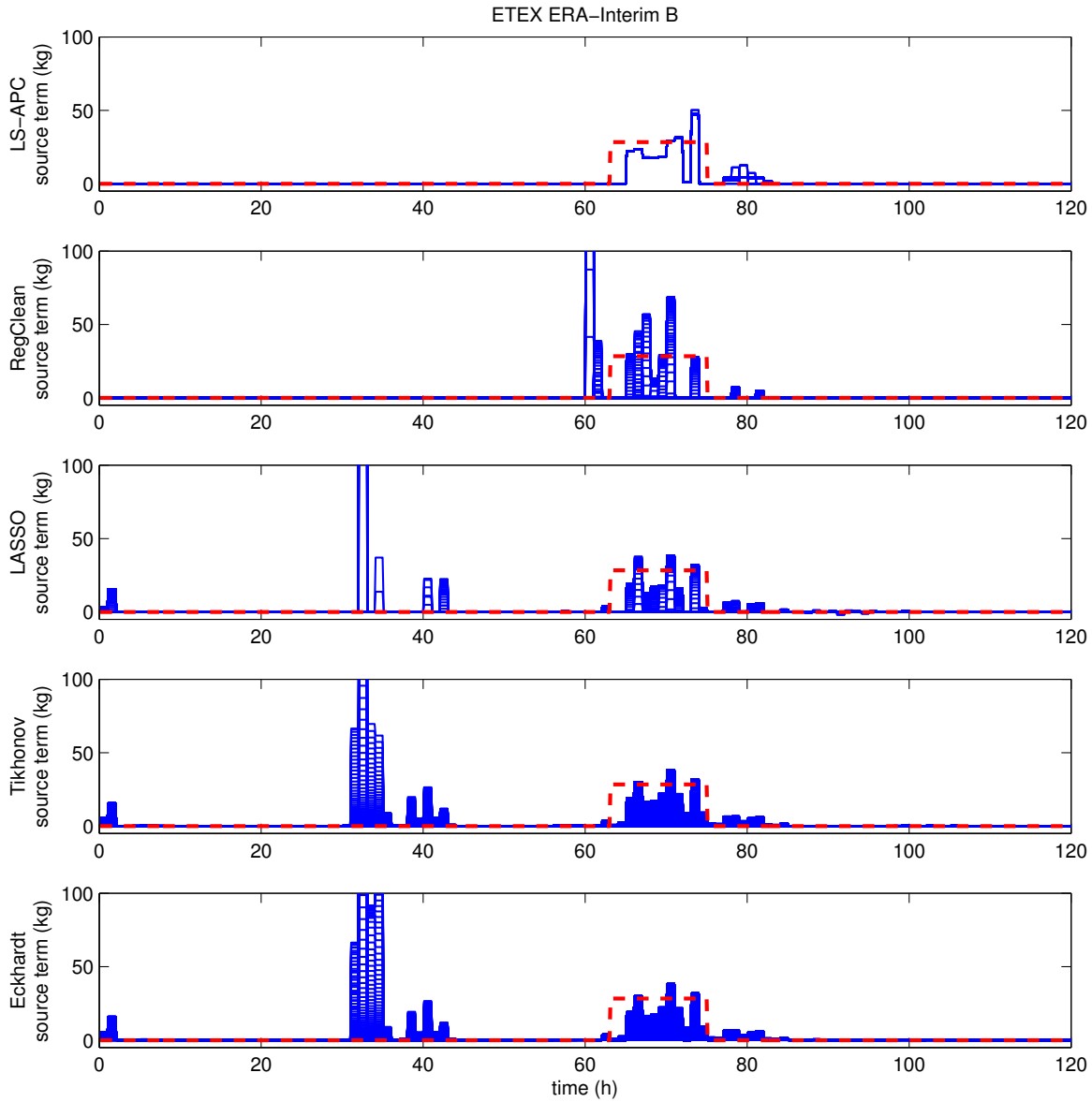

**Figure 7.** Comparison of the estimated source term for data ETEX ERA-Interim B for all settings (221 values) of the tuning parameter $\alpha$ using all algorithms. For LS-APC, all estimates are overlapping, for algorithms sensitive to this choice, lines for different value of the tuning parameter are plotted next to each other forming an area. The true source term is denoted by the dashed red line.

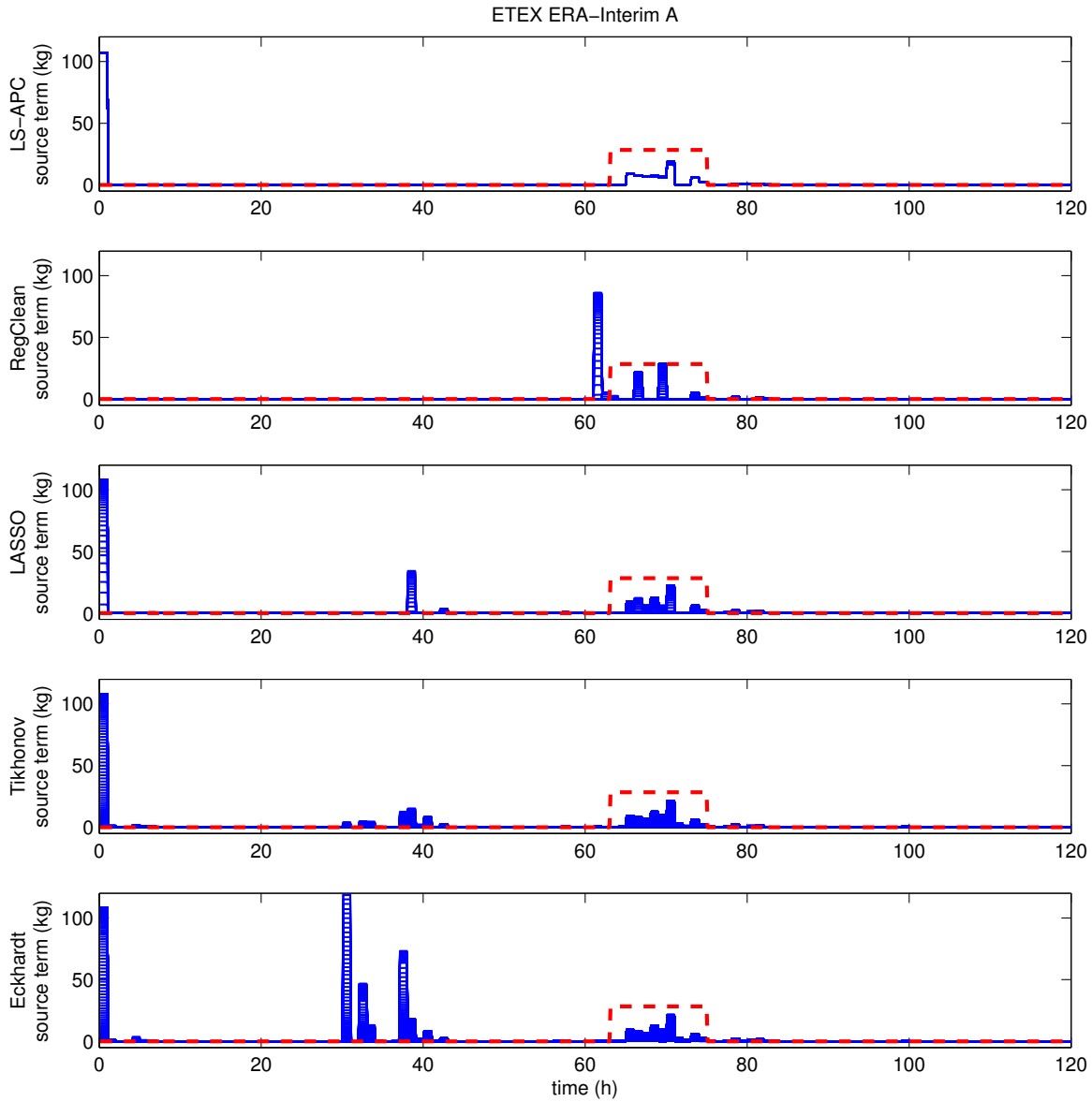

**Figure 8.** Comparison of the estimated source term for data ETEX ERA-Interim A for all settings (221 values) of the tuning parameter $\alpha$ using all algorithms. For LS-APC, all estimates are overlapping, for algorithms sensitive to this choice, the lines for different value of the tuning parameter are plotted next to each other forming an area. The true source term is denoted by the dashed red line.