# Peer review of "LS-APC v1.0: A tuning-free method for the linear inverse problem and its application to source term determination"

_Geoscientific Model Development, 2016_

## Short Comment (SC1) · 30 Mar 2016

Dear authors,

In my role as Executive editor of GMD, I would like to bring to your attention our Editorial version 1.1:

http://www.geosci-model-dev.net/8/3487/2015/gmd-8-3487-2015.html

This highlights some requirements of papers published in GMD, which is also available on the GMD website in the 'Manuscript Types' section:

http://www.geoscientific-model-development.net/submission/manuscript_types.html

In particular, please note that for your paper, the following requirement has not been met in the Discussions paper:

[Figure]

- "The main paper must give the model name and version number (or other unique identifier) in the title."

Please add the name (LS-ADP ? ) and a version number for your algorithm in the title upon your revised submission to GMD.

Yours,

Astrid Kerkweg
* * *

---

## Referee Comment (RC1) · Anonymous Referee #1 · 29 Jul 2016

This paper provides an interesting description of a Variational Bayesian approach to source term estimation that allows for the tuning of the hyper-parameters to be performed, based on the information content of the measurements. The paper is a valuable contribution, in that the optimization of the uncertainty hyper-parameters does not require pre-specified or pre-optimized uncertainties, as is often the case in Bayesian inversions. However, there are a number of issues that should be addressed before the paper is ready for publication. The authors have omitted to mention a range of studies in the literature that have previously addressed the problem of objectively defining these hyper-parameters, and how this work compares to those that have gone before. In addition, the paper is hindered by a lack of explanation in places, making it occasionally difficult to follow. A more thorough description of how this work compares to other hyper-parameter estimation approaches is required, along with the remedying of

other issues outlined below, in order for a more polished manuscript to be produced.

Specific Comments:

1. Page 1, Line 24: "...this two-pronged approach..." What exactly is meant by this? Top-down inversion studies are normally performed independently of the compilation of bottom-up inventory studies.

2. Page 2, Line 4: Could the authors be more specific as to what "other bottom-up information" entails?

3. Page 2, Lines 20-23: The authors have neglected to mention that many studies do not select these tuning parameters subjectively, and there have been a number of studies that have defined objective criteria for this purpose. For known location source-term estimation, examples include Davoine and Bocquet (2007) or Winiarek et al. (2012). In trace gas inversions Michalak et al. (2005) optimized covariance parameters using maximum likelihood estimation, and a similar approach was used in Berchet et al. (2013). In a perhaps more closely related approach to variational Bayes, Ganesan et al. (2014) used an MCMC method to estimate the hyper-parameters using the data. A discussion of these other approaches is needed, in order to contextualise this work.

4. Page 3, Line 7 & 10: The use of the term "State of the art methodology" is a bit of a push. The work of Eckhardt et al. (2008) may provide a useful reference to compare to, but there have been many examples of advances beyond the use of subjectively prescribed uncertainties since then (if not before, see above).

5. Page 4, Algorithm 1: The term "stopcond" needs explaining, or some reference made to it in the text.

6. Page 4, Line 4: What is meant by the "potential prior mean" and why is this subtracted from both sides of the equation?

7. Page 4, Line 11: "The method of Eckhardt et al. (2008) has Bayesian interpretation

as a maximum a posteriori probability estimate of the following model:" This sentence did not make sense to me, please clarify or rephrase.

8. Page 5, Line 8: Could the Variational Bayes approach also be extended to deal with this second problem?

9. Page 5, Line 10: "Approximate inference of these values does not yield acceptable results". This statement is too vague, please expand. Are the authors referring to MCMC approximations, and if so why would these be unacceptable? As I understand, the advantage of Variational Bayes over MCMC is mostly a matter of speed, but Variational Bayes may be more susceptible to bias. Perhaps this could be commented on.

10. Page 5. Line 22: "The selection of these constants will be discussed later in this paper." It would be helpful to point the reader to the exact section. As it stands, I am not certain that any discussion on the selection of these constants actually appears in the text.

11. Page 6, Lines 15-16: Given that the authors state that the expected values of l_j are either 0 or -1, is there a need for such an uninformative range on l_j (-1 +/- 100)? What would be the effect of a smaller range on l_j? Similarly what would the effect of further relaxation be, and why is this not recommended?

12. Page 7, Algorithm 2, 2 (c) and (d): It is unclear which equations in Appendix B define the covariance structure. I assume it is Eq. (B1), but this could be made more obvious.

13. Page 8, Lines 6-7: How much higher is the computational cost expected to be?

14. Page 9, Lines 29-32: In A) what is the Lagrangian timescale? It would be helpful to explain why different results are expected for different time steps, and what uncertainty running two different time steps might account for. It was a little surprising to find no subsequent discussion of the differences between configurations A and B. Why, for

instance, is the artefact in ERA-Interim A not seen in ERA-interim B?

15. Page 10, Lines 13-14: For the avoidance of doubt please make clear which of 230 kg and 340 kg is the posterior and which is the true source term. Furthermore, there surely must be some uncertainty on the posterior source term? "Quite similar" is a vague description, and may not be entirely accurate given one term is 50% larger than the other. Could the authors comment on whether the results are statistically similar?

16. Page 11, Lines 3-4: What do the top rows in Figures 5 & 6 show? It is not immediately obvious what the graphs are displaying, and it would help to explain this in the text. These graphs need more explaining both in the text and the figure caption.

17. Page 11, Lines 18-20: I assume the range of possible solutions is shown by the blue fill, but this should be made explicit in the text and the figure captions.

18. Page 11, Lines 21-23: How long does it take to run, and how much more expensive is it than the simpler techniques? How would the computational cost scale with the dimension of both the parameters and data vectors?

19. Figure 5 & 6: Which tuning parameter does the x-axis refer to and does it have a unit? Shouldn't the y-axis in the top panel also have units?

Technical Corrections:

20. Throughout the manuscript I believe equations should be referenced as "Eq. (1)". Similarly figures should appear as "Fig. 1". Details of GMD guidelines can be found here: http://www.geoscientific-model-development.net/for_authors/manuscript_preparation.html

21. Page 9, Line 13: Figure 4 is referenced before figure 3

22. Figure 2: The x-axes appear to be missing a label. The dotted lines also appear very faint and hard to see.

23. Figure 3: No x-axis label

References

Berchet, A., Pison, I., Chevallier, F., Bousquet, P., Conil, S., Geever, M., Laurila, T., Lavric, J., Lopez, M., Moncrieff, J., Necki, J., Ramonet, M., Schmidt, M., Steinbacher, M., and Tarniewicz, J.: Towards better error statistics for atmospheric inversions of methane surface fluxes, Atmos Chem Phys, 13, 7115-7132, 2013.

Davoine, X. and Bocquet, M.: Inverse modelling-based reconstruction of the Chernobyl source term available for long-range transport, Atmos Chem Phys, 7, 1549-1564, 2007.

Ganesan, A. L., Rigby, M., Zammit-Mangion, A., Manning, A. J., Prinn, R. G., Fraser, P. J., Harth, C. M., Kim, K. R., Krummel, P. B., Li, S., Mühle, J., O'Doherty, S. J., Park, S., Salameh, P. K., Steele, L. P., and Weiss, R. F.: Characterization of uncertainties in atmospheric trace gas inversions using hierarchical Bayesian methods, Atmos. Chem. Phys., 14, 3855-3864, 2014.

Michalak, A. M., Hirsch, A., Bruhwiler, L., Gurney, K. R., Peters, W., and Tans, P. P.: Maximum likelihood estimation of covariance parameters for Bayesian atmospheric trace gas surface flux inversions, J Geophys Res-Atmos, 110, 2005.

Winiarek, V., Bocquet, M., Saunier, O., and Mathieu, A.: Estimation of errors in the inverse modeling of accidental release of atmospheric pollutant: Application to the reconstruction of the cesium-137 and iodine-131 source terms from the Fukushima Daiichi power plant, J Geophys Res-Atmos, 117, 2012.
* * *

---

## Referee Comment (RC2) · Anonymous Referee #2 · 14 Aug 2016

General comments

In the paper the authors propose to apply the Variational Bayesian methodology to estimate the tuning parameters of the objective function given in Eckhardt et al. (2008). The authors describe the method and the algorithm to compute such tuning parameters. Then they show the performance of the proposed algorithm using a synthetic dataset, and the ETEX dataset. Its performance in the ETEX dataset is compared with the performance of other state-of-the art algorithms.

Individual questions / issues

- page 3, line 20: For clarity, the optimization problem can be written including the non-negative constrain for x

x = argmin_x (J1 + J2 + J3) s.t. x >= 0

- page 4, line 16: The Gaussian assumption is a good choice if, in fact, the errors in the model are Gaussian. Otherwise, this can cause deviations in the estimation. The authors should justify why the Gaussian assumption is a reasonable one in this case. Also, this particular regularization enforces smoothness in the solution. It should be mentioned that it is not suitable for releases generated, for example, during explosions.

- page 5, line 1: gamma(x) should be defined more precisely.

- page 5, line 10: define approximate inference. The authors also should explain how they concluded that this method does not yield acceptable results.

- page 5, line 16: why do the authors assume that the variance for all the measurements is the same? Is it not more reasonable to define w as a vector instead of a scalar?

- page 5, line 20: the authors should explain why the gamma distribution is chosen to model w.

- page 5, line 25: explain in more detail why this particular relaxation has been chosen.

- page 6, line 16: The authors should explain why they conclude that a wider range for that parameter l_j is not recommended. What are the effects if the range is wider?

- page 6, line 23: does conditional independence make sense here? The authors should explain why they are making that assumption.

- page 7, line 11: the derivation of the parameters is not in the Appendix B. Only the definition of the parameters is given. An explanation on how the authors arrive there is recommended.

- page 8, line 1: since local minima exist, good initial points should be taken, or several initial points may be considered.

- page 8, line 8: The authors should also comment on the convergence guarantees of

the algorithm.

- algorithm 2: Step 2 is not clear. Is <x>ˆ(i) equal to <x>? What is exactly the analytic expression for <x>ˆ(i)?

- page 8 , line 12: The authors should provide the condition number of the matrix M.

- Figure 2: x-axis labels are missing

- page 8, line 14. It is not clear if the three 'sets' refer to three different synthetic experiments, or not.

- page 8 : in the experiments with the synthetic dataset, it would be interesting to compare the estimated parameters w.r.t their truth value, i.e. as in the case for the source term, a red line representing the ground truth could be added in the other plots. It will give a more precise idea of the quality of the estimate of the parameters.

- Matlab code: the Matlab code provided reproduces the results given by the LS APC algorithm. To facilitate the reproducibility of all the results in the paper, the authors should include the code used to generate the results given by the other algorithms as well.
* * *

---

## Author Comment (AC1) · 9 Sep 2016

**Please add the name (LS-ADP ? ) and a version number for your algorithm in the title upon your revised submission to GMD.**

Thank you for this note, we added the name and the version of the model (LS-APC v1.0) into the title of the article.
* * *

---

## Author Comment (AC2) · 9 Sep 2016

We would like to thank you for providing us with detailed reviews of our paper. We have considered all the comments and notes and we are glad that we can submit a revised version of our paper. In the following text, we will respond to all comments.

**This paper provides an interesting description of a Variational Bayesian approach to source term estimation that allows for the tuning of the hyper-parameters to be performed, based on the information content of the measurements. The paper is a valuable contribution, in that the optimization of the uncertainty hyper-parameters does not require pre-specified or pre-optimized uncertainties, as is often the case in Bayesian inversions. However, there are a number of issues that should be addressed before the paper is ready for publication. The**

**authors have omitted to mention a range of studies in the literature that have previously addressed the problem of objectively defining these hyper-parameters, and how this work compares to those that have gone before. In addition, the paper is hindered by a lack of explanation in places, making it occasionally difficult to follow. A more thorough description of how this work compares to other hyper-parameter estimation approaches is required, along with the remedying of other issues outlined below, in order for a more polished manuscript to be produced.**

**1  Specific Comments:**

**1. Page 1, Line 24: ". . .this two-pronged approach. . ." What exactly is meant by this? Top-down inversion studies are normally performed independently of the compilation of bottom-up inventory studies.**

We agree to this and have extended this sentence to: For determining the emissions of greenhouse gases into the atmosphere, such an approach has become very common. In particular, total greenhouse gas emissions are the result of both anthropogenic and natural emissions. Bottom-up inventories for anthropogenic emissions should, at least in principle, be quite accurate but a verification using top-down methods is desirable (Stohl et al., 2009; Bergamaschi et al., 2015). Natural emissions are often poorly constrained with bottom-up methods and thus the role of top-down methods is even more important (Tans et al., 1990; Rayner et al., 1999).

**2. Page 2, Line 4: Could the authors be more specific as to what "other bottom-up information" entails?**

We agree that this formulation was a bit vague and have replaced "other bottom-up information can be very incomplete or ..." with "information on the magnitude of the emissions, their temporal variations and, occasionally, the emission altitude, can be

very incomplete or ...."

**3. Page 2, Lines 20-23: The authors have neglected to mention that many studies do not select these tuning parameters subjectively, and there have been a number of studies that have defined objective criteria for this purpose. For known location source-term estimation, examples include Davoine and Bocquet (2007) or Winiarek et al. (2012). In trace gas inversions Michalak et al. (2005) optimized covariance parameters using maximum likelihood estimation, and a similar approach was used in Berchet et al. (2013). In a perhaps more closely related approach to variational Bayes, Ganesan et al. (2014) used an MCMC method to estimate the hyper-parameters using the data. A discussion of these other approaches is needed, in order to contextualise this work.**

Thank you very much for this valuable comment. We studied the recommended papers and their followups. All papers are very relevant and we extended the introduction of our paper. We added discussion on modeling of covariance parameters from the application point of view to the third paragraph. We also added the word "statistical" the the next paragraph the emphasize that the forth paragraph is review of statistical literature on the same topic.

**4. Page 3, Line 7 & 10: The use of the term "State of the art methodology" is a bit of a push. The work of Eckhardt et al. (2008) may provide a useful reference to compare to, but there have been many examples of advances beyond the use of subjectively prescribed uncertainties since then (if not before, see above).**

We agree. What we meant here is not that Eckhardt et al. (2008) have developed the most advanced method, but rather that it is a typical example of inverse modeling in the atmospheric sciences. We have replaced the term with "standard methodology".

**5. Page 4, Algorithm 1: The term "stopcond" needs explaining, or some reference made to it in the text.**

First of all, we replaced the term "stopcond" by the symbol $\delta$ for consistency (parameters using Greek symbols). We also added the comment on this stopping condition the the text.

**6. Page 4, Line 4: What is meant by the "potential prior mean" and why is this subtracted from both sides of the equation?**

Wee agree that this formulation was misleading and we reformulated it completely. Here, we referred only to a technical step (from (Eckhardt et al., 2008), Eq. (6) and (7)) where prior knowledge on source term, $\mathbf{x}^a$, is included via change of coordinates $M(\mathbf{x} - \mathbf{x}^a) = M\tilde{\mathbf{x}}$. We now give a more general change of coordinates that can also accommodate for known covariance matrix of observations.

**7. Page 4, Line 11: "The method of Eckhardt et al. (2008) has Bayesian interpretation as a maximum a posteriori probability estimate of the following model:" This sentence did not make sense to me, please clarify or rephrase.**

For a loss function used in an optimization based inference method, it is often possible to find a statistical model that has logarithm equal to that loss function. It is certainly the case with the method of Eckhardt et. al. We rephrased the introduction and extended description of the positivity enforcement using new equation (5), which was added in reaction to request of Rev. 2. We hope that this new formulation is clearer.

**8. Page 5, Line 8: Could the Variational Bayes approach also be extended to deal with this second problem?**

Indeed, it could be extended to the problem of model $M$ selection. However, the extension is not trivial and is beyond the scope of this paper. We also commented this in the revised paper.

**9. Page 5, Line 10: "Approximate inference of these values does not yield acceptable results". This statement is too vague, please expand. Are the authors referring to MCMC approximations, and if so why would these be unacceptable?**

**As I understand, the advantage of Variational Bayes over MCMC is mostly a matter of speed, but Variational Bayes may be more susceptible to bias. Perhaps this could be commented on.**

We agree that this formulation was too vague and we reformulated it. For Variational Bayes, there is a computational problem in analytical solution of the determinant of the covariance matrix. However, it is true that it can be overcome by MCMC. Probably more important reason for our choice was the ability of the prior to model abrupt changes. We reformulated the whole paragraph.

**10. Page 5. Line 22: "The selection of these constants will be discussed later in this paper." It would be helpful to point the reader to the exact section. As it stands, I am not certain that any discussion on the selection of these constants actually appears in the text.**

Indeed, there was little discussion on the choice of these parameters. The only requirement we put on these is non-informativeness of the prior and therefore negligible impact on the results. These were chosen as $10^{-10}$ in Algorithm 2. The whole sentence was rephrased.

**11. Page 6, Lines 15-16: Given that the authors state that the expected values of l_j are either 0 or -1, is there a need for such an uninformative range on l_j (-1 +/- 100)? What would be the effect of a smaller range on l_j? Similarly what would the effect of further relaxation be, and why is this not recommended?**

Once again, we aim at as non-informative choice as possible. Value $\pm100$ was chosen experimentally as a compromise between non-informativeness and robustness/stability of the methods. Discussion on these prior constants was added to the paper.

**12. Page 7, Algorithm 2, 2 (c) and (d): It is unclear which equations in Appendix B define the covariance structure. I assume it is Eq. (B1), but this could be made more obvious.**

[Figure]

We added the explicit referencing into the Algorithm 2 in order to clarify it.

**13. Page 8, Lines 6-7: How much higher is the computational cost expected to be?**

Exact increase of the computational cost is hard to evaluate. The dominating operation is inverse of the covariance matrix which needs to be evaluated, hence, it scales with circa $O(n^{2.4})$. We added this note into the paper.

**14. Page 9, Lines 29-32: In A) what is the Lagrangian timescale? It would be helpful to explain why different results are expected for different time steps, and what uncertainty running two different time steps might account for. It was a little surprising to find no subsequent discussion of the differences between configurations A and B. Why, for instance, is the artefact in ERA-Interim A not seen in ERA-interim B?**

We added more description for the FLEXPART runs. It is difficult to explain exactly why simulation results are different with configurations A and B, as only configuration A is physically correct. It is expected, however, that configuration B leads to systematically slightly smaller concentrations as the density differences in the boundary layer are ignored with this option. At individual stations (especially stations close to the source) larger differences can occur simply due to the inaccurate treatment of turbulent dispersion in configuration B. This also depends on the meteorological data input, and so it is not surprising to see larger differences with one meteorological data set than with the other.

**15. Page 10, Lines 13-14: For the avoidance of doubt please make clear which of 230 kg and 340 kg is the posterior and which is the true source term. Furthermore, there surely must be some uncertainty on the posterior source term? "Quite similar" is a vague description, and may not be entirely accurate given one term is 50% larger than the other. Could the authors comment on whether the results are statistically similar?**

Indeed, there is uncertainty in the estimated source term which can be quantified using our algorithm. We added the uncertainty bounds into the Fig. 4 where the 99% highest posterior density region is shown using gray fill region. We also agree that the statement "quite similar" was vague and we reformulated this in the paper. We now compare the total true source term with highest posterior density region. However, we are aware that statistical significance of this results is still questionable since the uncertainty in $M$ is not fully quantified.

**16. Page 11, Lines 3-4: What do the top rows in Figures 5 & 6 show? It is not immediately obvious what the graphs are displaying, and it would help to explain this in the text. These graphs need more explaining both in the text and the figure caption.**

We added the description to these graphs into the text and added a references into the captions. Since the algorithms rely on tunning parameter $\alpha$, we computed the source term for $\alpha \in < e^{-15}, e^{+7} >$ using each algorithm and then computed the mean absolute errors between computed source terms and the true source term.

**17. Page 11, Lines 18-20: I assume the range of possible solutions is shown by the blue fill, but this should be made explicit in the text and the figure captions.**

There was a mistake in the text in description of these figures and we appreciate that you pointed out this. We clarify this in the text as well as in captions of figures.

**18. Page 11, Lines 21-23: How long does it take to run, and how much more expensive is it than the simpler techniques? How would the computational cost scale with the dimension of both the parameters and data vectors?**

We added comments on computational cost to the paper. Specifically, we added comment to the discussion on the LS-APC algorithm in Sec. 3 and further discussion on computational cost on ETEX experiment at the end of Sec. 5.

**19. Figure 5 & 6: Which tuning parameter does the x-axis refer to and does it**

[Figure]

**have a unit? Shouldn't the y-axis in the top panel also have units?**

We refer here to the tunning parameter $\alpha$. We clarified this by adding the symbol $\alpha$ to labels of the x-axis. This parameter is dimensionless. Indeed, the y-axis in the top panel have units (kg) and we added the units to the label of the axis.

**2   Technical Corrections:**

**20.   Throughout the manuscript I believe equations should be referenced as "Eq.   (1)".   Similarly figures should appear as "Fig.   1".   Details of GMD guidelines can be found here:   http://www.geoscientific-modeldevelopment.net/for_authors/manuscript_preparation.html**

Thank you for this note, we corrected the referencing through the whole paper in order to meet the house standard.

**21. Page 9, Line 13: Figure 4 is referenced before figure 3**

This mistake was made during finalizing the manuscript and is corrected now.

**22.  Figure 2: The x-axes appear to be missing a label.  The dotted lines also appear very faint and hard to see.**

We added labels "Time step" to the x-axis. Since it is simulated example, there is no need for units. We also replaced the doted lines by dashed black lines which should be easier to recognize.

**23. Figure 3: No x-axis label**

We added the x-axis labels.

---

## Author Comment (AC3) · 9 Sep 2016

We would like to thank you for providing us with detailed reviews of our paper. We have considered all the comments and notes and we are glad that we can submit a revised version of our paper. In the following text, we will respond to all comments.

**In the paper the authors propose to apply the Variational Bayesian methodology to estimate the tuning parameters of the objective function given in Eckhardt et al. (2008). The authors describe the method and the algorithm to compute such tuning parameters. Then they show the performance of the proposed algorithm using a synthetic dataset, and the ETEX dataset. Its performance in the ETEX dataset is compared with the performance of other state-of-the art algorithms.**

[Figure]

**1   Specific Comments:**

**1. page 3, line 20: For clarity, the optimization problem can be written including the non-negative constrain for x, x = argmin_x (J1 + J2 + J3) s.t. x >= 0**

We agree that explicit statement is easier to follow and added a new equation (5) to emphasize it. It also makes comparison with the probabilistic approach easier.

**2. page 4, line 16: The Gaussian assumption is a good choice if, in fact, the errors in the model are Gaussian. Otherwise, this can cause deviations in the estimation. The authors should justify why the Gaussian assumption is a reasonable one in this case. Also, this particular regularization enforces smoothness in the solution. It should be mentioned that it is not suitable for releases generated, for example, during explosions**

Thanks for this suggestion. Indeed, suitability of the prior to abrupt changes was one of our primary motivation. It is now explicitly mentioned in the text. It is hard to justify the assumption of Gaussian noise since the residue is of complex nature here. We have added a sentence that this choice is motivated by tractability of its inference.

**3. page 5, line 1: gamma(x) should be defined more precisely.**

The $\gamma$ term is now defined in text as logarithm of the characteristic function. Since this is only a minor illustrative point, we wanted to avoid the use of formal mathematical approach. The main point is that (9) and (5) are equivalent problems, which is now explicitly stated.

**4. page 5, line 16: why do the authors assume that the variance for all the measurements is the same? Is it not more reasonable to define w as a vector instead of a scalar?**

It is important to distinguish if the covariance matrix of the observations is known or not. If it is known, it can be used to transform the problem into isotropic noise. The

explicit equations for this transformation have been added to the paper.

A more demanding case is if we assume that the observation variance is unknown. Assuming completely independent variance for each observation leads to over-parametrization which can be addressed only by introduction of further restrictive assumptions. It is an interesting topic of future research, however, in our experiments the assumption of same variance was found to be quite reasonable and robust choice.

**5. page 5, line 20: the authors should explain why the gamma distribution is chosen to model w.**

The gamma distribution was chosen due to conjugacy to the Gaussian distribution so the prior and the posterior distribution have the same form in the Variational Bayes procedure which is beneficial for computational reasons. We commented this in the text a we added the citation for this model.

**6. page 5, line 25: explain in more detail why this particular relaxation has been chosen.**

This relaxation was made the preserve the tri-diagonality of the matrix $\Sigma_{\mathbf{x}}$ and using this model, each diagonal can be modeled separately. We added the comment into the text.

**7. page 6, line 16: The authors should explain why they conclude that a wider range for that parameter l_j is not recommended. What are the effects if the range is wider?**

We agree that this formulation need to be specified. We reformulated the paragraph and discussion on these prior constants was added to the paper.

**8. page 6, line 23: does conditional independence make sense here? The authors should explain why they are making that assumption.**

Indeed, this assumption seems arbitrary. Its motivation is primarily simple solution of

the implied variational problem. However, experience indicate that estimation of linear models under this assumption yields results very close to much more expensive MCMC approaches. We added few references to relevant literature.

**9. page 7, line 11: the derivation of the parameters is not in the Appendix B. Only the definition of the parameters is given. An explanation on how the authors arrive there is recommended.**

Indeed, teh parameers are not derived in Appendix B. Derivation of the parameters is long and routine procedure. We have now added reference to a book where it is described and replaced he word derived by 'given'.

**10. page 8, line 1: since local minima exist, good initial points should be taken, or several initial points may be considered.**

Indeed, the initialization of the LS-APC algorithm needs to be selected. We extended description of the proposed initialization in Algorithm 2. We have good experience with this choice in our experiments, however, it is not the only possibility. Much more advanced search strategies for global solution using any global optimization method are possible.

**11. page 8, line 8: The authors should also comment on the convergence guarantees of the algorithm.**

Actually, the VB algorithm converges only to local extreme as it is now cited in the text.

**12. algorithm 2: Step 2 is not clear. Is <x>ˆ(i) equal to <x>? What is exactly the analytic expression for <x>ˆ(i)?**

Thank you for this remark, the description of the iteration indexing $<\mathbf{x}>^{(i)}$ is certainly needed. We clarify this when introducing the initialization of the algorithm. All required moments are given in Appendix A and detailed references to particular equations have been added.

**13. page 8 , line 12: The authors should provide the condition number of the matrix M.**

We agree and we added the condition number of the matrix $M$ into the text. Since the matrix is not ill conditioned, we also reformulated the sentence.

**14. Figure 2: x-axis labels are missing**

The labels are added now.

**15. page 8, line 14. It is not clear if the three 'sets' refer to three different synthetic experiments, or not.**

In the synthetic experiment, the same matrix $M$ was used. The difference is in the realization of noise, i.e. in vector $\mathbf{y}$. We added a comment into the description of the data.

**16. page 8 : in the experiments with the synthetic dataset, it would be interesting to compare the estimated parameters w.r.t their truth value, i.e. as in the case for the source term, a red line representing the ground truth could be added in the other plots. It will give a more precise idea of the quality of the estimate of the parameters.**

Derivation of the ground truth for these parameter is not very clear. First, there is more ways how to define the "true prior covariance matrix". One possible choice is the empirical covariance from the single realization of the parameter. Then, the matrix has rank=1 and decomposition into choleski factors is not unique. There is a number of matrices giving the same covariance. Probably the most illustrative example could be display of the implied covariance as an image, such as in Fig. 1, bottom row. However, we are not sure that it would be understandable.

**17. Matlab code: the Matlab code provided reproduces the results given by the LS APC algorithm. To facilitate the reproducibility of all the results in the paper, the authors should include the code used to generate the results given by the**

**other algorithms as well.**

We agree that it will be beneficial to include also other algorithms into the code posted online. We include our implemnetation of the LASSO and Tikhonov algorithm into the MATLAB package provided online. Evaluation of the remaining methods was done using their code. The RegClean algorithm by Martinez-Camara et al., (2014), can be downloaded as a supplement to their paper (published in GMD). The algorithm from Eckhardt et al., (2008), is not implemented in MATLAB (see Eckhardt paper for details) so it would not be consistent to provide it together with the LS-APC algorithm. Therefore, we are not redistributing these algorithms.

[Figure]

Interactive
comment

**Fig. 1.**

---

## Author Response (AR2)

**Response to reviewers**

November 8, 2016

Dear editor and reviewers,

thank you for your response and suggestions. We have corrected all mentioned typos in the manuscript.

Yours Sincerely,

Vaclav Smidl, Ondrej Tichy, Radek Hofman and Andreas Stohl